# Biomass Nanoporous Carbon-Supported Pd Catalysts for Partial Hydrogenation of Biodiesel: Effects of Surface Chemistry on Pd Particle Size and Catalytic Performance

**DOI:** 10.3390/nano11061431

**Published:** 2021-05-28

**Authors:** Parncheewa Udomsap, Sirasit Meesiri, Nuwong Chollacoop, Apiluck Eiad-Ua

**Affiliations:** 1College of Materials Innovation and Technology, King Mongkut’s Institute of Technology Ladkrabang, Chalongkrung Rd., Ladkrabang, Bangkok 10520, Thailand; parncheewa.udo@entec.or.th; 2School of Energy and Environment, University of Phayao, Phaholyothin Rd., Mae Ka, Mueang Phayao, Phayao 56000, Thailand; sirasit.khing11@gmail.com; 3Energy Innovation Research Group, National Energy Technology Center, 114 Thailand Science Park, Phaholyothin Rd., Klong 1, Klong Luang, Pathumthani 12120, Thailand; nuwong.cho@entec.or.th

**Keywords:** nanoporous carbon, chemical activation, surface chemistry, Pd catalyst, partial hydrogenation, particle size effect

## Abstract

Two types of cattail flower-derived nanoporous carbon (NPC), i.e., NPC activated with KOH and H_3_PO_4_, were produced and characterized using several techniques (e.g., Raman spectroscopy, nitrogen adsorption, and X-ray photoelectron spectroscopy). The influence of the carbon support characteristics on the particle sizes and chemical states of Pd in the synthesized Pd/NPC catalysts, which affect the catalytic activity and product selectivity, was analyzed. The surface chemistry properties of NPC were the main factors influencing the Pd particle size; by contrast, the textural properties did not significantly affect the size of the Pd particles on NPC supports. The use of Pd nanoparticles supported on the rich-functionalized surface carbons obtained by H_3_PO_4_ activation led to superior catalytic activity for the polyunsaturated fatty acid methyl ester (poly-FAME) hydrogenation, which could achieve 90% poly-FAME conversion and 84% selectivity towards monounsaturated FAME after a 45-min reaction time. This is due to the small Pd nanoparticle size and the high acidity of the catalysts, which are beneficial for the partial hydrogenation of poly-FAME in biodiesel. Conversely, the Pd nanoparticles supported on the high-surface-area carbon by KOH activation, with large Pd particle size and low acidity, required a longer reaction time to reach similar conversion and product selectivity levels.

## 1. Introduction

The use of nanoporous carbon (NPC) as a support material for precious metals is significantly increasing, owing to its unique characteristics: its inertness in both acidic and basic media, its ability to delay active phase sintering, and the ability to recover precious metals from the spent catalysts by burning the carbon off. In addition, the pore size distribution and surface chemistry properties can be modified according to the prospective application. A support material for a metal catalyst provides the dispersion and stability of metal particles on the surface. The supported catalysts represent a higher surface area of active catalytic phases compared to the bulk metal [1,2,3,4]. Therefore, NPC materials are attractive candidates to be supporting materials for the preparation of heterogeneous catalysts. The catalytic properties of the supported catalysts are mainly influenced by the texture and surface chemistry of the NPC support, affecting the activity and selectivity of the catalyst [5,6,7,8].

The pore structure and surface chemistry of NPC depend mainly on the nature of the material feedstocks and the activation process (physical or chemical) [9,10]. Owing to the low cost, sustainability, and renewable precursor, biomass (natural or agricultural waste) has been used for the production of NPC materials. The cattail flower (CTF) is an abundantly available weed in Thailand and has no economic value. Conversion of CTF biomass into NPC material would increase the value of the land and also give unwanted flora a positive environmental impact. In addition, the typical chemical composition of the CTF gives it the potential to be a raw material for carbon production [10,11].

In the preparation of an NPC material, two basic steps are performed. The first step is carbonization and the second is activation. In the carbonization stage, the carbon content of the carbonaceous precursor is increased by the removal of the volatile matter. In general, hydrothermal carbonization (HTC) is used to convert the wet and low bulk density precursors into hydrochar through thermochemical degradation [12]. According to the physical properties of CTF biomass, which is a short fiber and low bulk density substance, they are quite difficult to activate during the further process. As a result, HTC is an appropriate process used earlier for increasing both the carbon content and the bulk density of the material prior to the activation stage.

The activation process is normally applied for enhancing the surface properties of hydrochar, which has a low adsorption ability, to make it a better material for many chemical reactions. In general, hydrochar can be activated through physical or chemical processes; however, the chemical process is preferable due to a significant development of the surface area and the higher carbon yield [11,13]. Pure microporous carbons or a mixture of micro- and mesoporous carbons can be obtained by varying the activating agent and the preparation conditions [14,15]. In addition, different activating agents can alter the nature and concentration of oxygen surface complexes, which commonly determine the acidic and basic carbon surface. Redkevich et al. [16] employed stone-fruit-activated carbon-containing oxygen and nitrogen species to prepare a Pd catalyst by wetness impregnation for hydrogen oxidation. They found that the Pd catalyst on activated carbon doped by diethylamine groups exhibited the highest Pd dispersion and the best hydrogen oxidation activity.

Depending on the type of activating agent used, the different biomass precursors react differently to give a variety of surface areas, pore volumes, surface chemistry, and yields. In general, the activation of biomass with a basic activating agent, such as KOH, produces carbon with very high surface areas of up to 2000 m^2^ g^−1^ [9]. The acid activating agent most commonly used is H_3_PO_4_, which gives high yields with a significant alteration in the surface properties and reactivity of the end carbon product [17]. Arami-Niya et al. studied the effects of the acidic activation (H_3_PO_4_) and neutral activation (ZnCl_2_) of palm shells on the textural properties of the corresponding activated carbons. The surface area and pore volume of the carbon product obtained with H_3_PO_4_ activation were reported to be higher than those of ZnCl_2_ activation [18]. ZnCl_2_ is, however, less preferable due to environmental problems and contamination of the carbon products, which is a concern, especially in pharmaceutical and food industries [17]. This is consistent with our previous work. Dechabun et al. [19] produced carbon with high porosity from Nipa palm husks through the HTC process at 180 °C, subsequently activating the sample with KOH at a temperature of 900 °C. Longprang et al. [20] employed cattail leaves for producing an NPC material by adopting HTC and activation using acid activating agents (HCl, HNO_3_, H_2_SO_4_, and H_3_PO_4_). They found that HTC at 200 °C for 12 h, followed by activation with H_3_PO_4_ at 900 °C, produced the NPC with the highest surface area (specific surface area ~624 m^2^ g^−1^), leading to increased dispersion of metal catalysts on the carbon support.

Supported Pd catalysts are usually used for the partial hydrogenation of polyunsaturated fatty acid methyl ester (poly-FAME) in biodiesel [21,22,23]. To promote the utilization of biodiesel fuel (BDF) in high blends, the biodiesel’s oxidation stability should be improved by decreasing the poly-FAME content to prevent the deterioration of fuel quality during storage [24]. Therefore, the partial hydrogenation of poly-FAME to monounsaturated FAME (mono-FAME) is a favored approach to substantially enhance the oxidation stability. Complete hydrogenation should be avoided to limit the saturated FAME content, which could lead to poor cold flow properties [25,26].

Herein, CTF biomass, which is abundant and low-cost, is used as a raw material to produce two types of NPC materials. Hydrothermal carbonization-assisted chemical activation with KOH and H_3_PO_4_ agents is applied for NPC synthesis. The potential of low structural and microporous home-made NPCs to be catalyst supports is investigated by preparing the corresponding NPC-supported Pd catalyst. The correlation of the textural properties and surface chemistry of these carbon supports with the particle size and chemical state of Pd in the supported Pd catalyst is thoroughly examined via transmission electron microscopy (TEM) and X-ray photoelectron spectroscopy (XPS). These results are affected by the catalytic activity and product selectivity of the catalysts for the partial hydrogenation of poly-FAME in commercial palm-oil-derived biodiesel (palm-BDF) to yield palm-BDF with correspondingly high oxidation stability.

## 2. Materials and Methods

### 2.1. Materials

CTF, as a biomass precursor for NPC production, was collected from the Ladkrabang area of Bangkok, Thailand. The brown fiber of the flower was peeled from the stalk and dried at 110 °C before the carbonization process. The proximate and elemental analyses of the raw CTF are shown in Table 1. Analytical-grade KOH and 85% H_3_PO_4_ (Merck, Darmstadt, Germany) were used as the activating reagents. Pd(NO_3_)_2_·*x*H_2_O (Alfa Aesar, Ward Hill, MA, USA), with a purity of ≥99.8 wt%, was used as received as a Pd precursor. The palm-BDF used as biodiesel feedstock for partial hydrogenation was obtained from Global Green Chemical Public Company Limited (Rayong, Thailand).

### 2.2. Preparation of CTF-Derived NPC Materials

Raw CTF was first carbonized by the hydrothermal method, which involved incubating a mixture of 30 g of CTF and 60 mL of deionized water in a Teflon-lined stainless-steel autoclave at 200 °C for 12 h. Afterwards, the reactor was rapidly cooled by immersion in water to stop the reaction. The solid product (hydrochar) was filtered and dried at 90 °C for 24 h. Subsequently, chemical activation was performed by soaking 50 g of the hydrochar in 200 mL of KOH or H_3_PO_4_ at an impregnating ratio of 1:2 (weight of hydrochar to activating reagent). Continuous mixing of the hydrochar with activating reagents for 4 h was followed by filtering and drying in an oven overnight at 110 °C. After that, the impregnated material was activated at a temperature of 900 °C with a heating rate of 5 °C min^−1^ under a N_2_ flow of 0.2 L min^−1^ and maintained for 2 h. After the activation process, the sample was rinsed several times with hot distilled water until the pH was neutral, and then it was filtered and dried at 110 °C. The obtained NPC samples were labeled NPC-K and NPC-H, where K and H indicate that KOH and H_3_PO_4_ were used as the respective activating agents.

### 2.3. The Preparation of Pd/NPC Catalysts

The prepared NPC was dried at 110 °C before Pd impregnation. Two Pd/NPC catalysts with Pd loadings of 5 wt% were prepared by incipient wetness impregnation using a palladium(II) nitrate [Pd(NO_3_)_2_·*x*H_2_O] aqueous solution. The Pd(NO_3_)_2_·*x*H_2_O precursor (0.0054 g) was dissolved in deionized water (0.14 mL). Then, the dried NPC supports were impregnated with a solution of Pd(NO_3_)_2_·*x*H_2_O under vacuum (approximately 50 kPa) at 25 °C and aged for 24 h. Subsequently, the impregnated catalyst was dried at 60 °C under vacuum 8 kPa for 6 h using a rotary evaporator and calcined in a horizontal tube furnace at 300 °C for 3 h with a ramping rate of 1 °C min^−1^ under 500 mL min^−1^ of a N_2_ flow. The catalysts were reduced at 300 °C for 1 h with a heating rate of 5 °C min^−1^ under a H_2_ flow of 50 mL min^−1^ before being applied in the partial hydrogenation reaction. The prepared catalysts were labeled Pd/NPC-K and Pd/NPC-H, respectively.

### 2.4. Characterizations

The raw CTF and carbon materials were characterized through proximate and elemental analyses, using a TGA/SDTA851 thermogravimetric analyzer (Mettler Toledo, Zurich, Switzerland) and a CHN628 elemental analyzer (LECO, St. Joseph, MI, USA). The surface morphology of the prepared carbon was analyzed by a scanning electron microscope (SEM), using a JSM-7800F (Prime) (JEOL, Tokyo, Japan) equipped with an energy-dispersive X-ray (EDX). The scanning was performed in situ on a sample powder using a 10 kV accelerating voltage. The EDX analysis was done to analyze the elemental composition. The transmission electron microscopy (TEM) analysis was done using a JEOL TEM-2100Plus with an accelerating voltage of 200 kV. The sample was sonicated to be suspended in ethanol and then dropped onto carbon-coated copper grids. The distribution of metal particle size was analyzed by ImageJ (National Institutes of Health (NIH), Bethesda, MD, USA) software. The phase structures of the samples were characterized by X-ray diffraction (XRD), conducted with a TTRAX III X-ray diffractometer (Rigaku, Tokyo, Japan) with a Cu Kα radiation source (λ = 0.15406 nm) in the 2θ range between 10° and 80° at a scanning rate of 0.08° s^−1^. Raman spectra were recorded using a LabRam HR evolution Raman spectrometer (Horiba, Kyoto, Japan) equipped with an LCX-532L diode-pumped solid-state laser (Oxxius, Lannion, France). The respective wavelength and laser spot size of the Ar laser were 532 nm and ~0.8 µm diameter. The textural properties of the prepared carbon and the corresponding catalysts were measured by N_2_ adsorption–desorption techniques at −196 °C using an ASAP 2460 surface area and porosity analyzer (Micromeritics, Norcross, GA, USA). The sample was pretreated at 150 °C under a vacuum system before the measurement. The specific surface area (S_BET_) was evaluated from the isotherm data using the Brunauer–Emmett–Teller (BET) model. The total pore volume (V_total_) was obtained from the accumulating volumes of adsorbed nitrogen at a relative pressure of 0.99. The micropore volume (V_micro_) was obtained with the cumulative pore volume using the nonlinear density function theory (NLDFT) method. The mesopore volume (V_meso_) was calculated by subtracting V_micro_ from V_total_. The average pore diameter (D_p_) was also analyzed using the NLDFT method. The functional groups on the carbon surface were analyzed by Fourier Transform Infrared Spectroscopy (FTIR), which was conducted on an IRTracer-100 (Shimadzu, Kyoto, Japan). Direct readings were conducted by attenuated total reflectance (ATR) in the 4000−400 cm^−1^ infrared region. The acidity of the samples was measured via the NH_3_ temperature-programmed desorption (NH_3_-TPD) using a BELCAT-B (BEL, Osaka, Japan) chemisorption analyzer. The samples (0.2 g) were pretreated with He (30 mL min^−1^) at 300 °C for 1 h. The sample was saturated with a 10% NH_3_/He mixture at 100 °C. Then, purging was done at 100 °C with He to eliminate the physically adsorbed NH_3_. The NH_3_-TPD was carried out by heating to a temperature range of 100–900 °C at a ramping rate of 5 °C min^−1^ under a He flow (30 mL min^−1^). The molecule desorption was monitored on-line by a BELMass (MicrotracBEL, Osaka, Japan) MS detector. The MS signals of NH_3_ were observed by the m/z ratio of 16. The pH at point zero charges (PZC) of the carbon materials was measured using the pH shift method [27]. The surface charge of the sample was determined by measuring the zeta potential of carbon-suspended solutions, varying their pH using a Zetasizer Nano ZS device, equipped with an MPT-2 titrator (Malvern Instruments, Malvern, UK). The solution’s pH was adjusted to the desired level by adding 0.1 M HCl or NaOH. The Pd loading was measured by an inductively coupled plasma atomic emission spectrometer (ICP-AES) using the ICPE-9820 equipped with a charge-coupled device (CCD) detector (Shimadzu, Kyoto, Japan). X-ray photoelectron spectroscopy (XPS) was applied to determine the chemical state of C, O, P, and Pd in the prepared carbons and catalysts using an AXIS Supra (Shimadzu-Kratos, Kyoto, Japan) XPS instrument. The measurements were performed in an ultrahigh vacuum system (1 × 10^−7^ Pa) with monochromatic Al Kα (hν = 1486.6 eV) radiation. The binding energy was calibrated by the C 1s peak (284.6 eV). The spectra were processed using ESCApe software (Shimadzu-Kratos, Kyoto, Japan).

### 2.5. Partial Hydrogenation of Poly-FAME

Partial hydrogenation of the palm-BDF to yield hydrogenated palm-BDF (H-FAME) was performed in a semi-batch type stainless steel reactor at 80 °C and 0.5 MPa of H_2_ under vigorous stirring. Firstly, 0.2 g of the reduced catalyst was added to a reactor containing 200 g of the palm-BDF. The reactor then eliminated the remaining air by purging with N_2_, followed by flowing of H_2_ and maintaining the H_2_ pressure at 0.5 MPa. The agitator was stirred at 700 rpm after the reaction temperature increased to 80 °C. The hydrogenated samples were collected to monitor the change in FAME composition during a 2-h reaction time. The compositions of FAME in biodiesel and the hydrogenated product were determined using a GC-2010 (Shimadzu, Kyoto, Japan) gas chromatography instrument, equipped with a flame ionization detector (FID). The HP-88 fused-silica capillary column (100 m × 0.25 mm × 0.2 µm) was used for compound separation. A 1-µL sample was injected into the oven at a 230 °C injector temperature with a split ratio of 75:1 under the He gas carrier. The vapor samples were initially separated at 140 °C; then, the temperature was increased to 240 °C at a heating rate of 4 °C min^−1^ and held for 15 min. The detector temperature was set at 250 °C. The FAME compositions were identified by comparing the retention time of reference compounds to the EN 14103 standard. The oxidative stability of the oil sample was determined using a 743 Rancimat (Metrohm, Herisau, Switzerland) instrument according to the EN 14112. The cloud point is one of the cold flow properties used to evaluate the wax precipitation potential of oil. It can be determined using a CPP 5GS (ISL PAC, Houston, TX, USA) automated cloud point and pour point analyzer according to the ASTM D2500 method.

## 3. Results and Discussion

### 3.1. Characteristics of the CTF-Derived NPC Materials

The proximate and elemental analyses of the NPC-K and NPC-H are presented in Table 1, in comparison to a raw CTF and the corresponding hydrochar as the starting biomass. Given the data of the proximate analysis, CTF presented a low moisture content (7.5%), high volatile matter (75.5%), an average content of fixed carbon (C_fixed_) (12.1%), and a low ash content (4.9%). The moderate C_fixed_ and high volatile contents of the CTF make this biomass a potential precursor for the production of NPC [10,11]. A low ash content is desired in carbon, representing the amount of inorganic material from the biomass precursor [28]. The elemental analysis revealed that raw CTF had 54 wt% of carbon, 37 wt% of oxygen, and 0.24 wt% of sulfur. The obtained hydrochar demonstrated an increase in C_fixed_ and the ash content after undergoing hydrothermal treatment to remove volatile matter, i.e., nitrogen, oxygen, and hydrogen, from CTF precursors through dehydration and decarboxylation, while the sulfur content was relatively similar to that of the raw CTF precursor.

After chemical activation, both NPC-K and NPC-H exhibited decreased volatile matter and ash contents, consequently increasing the C_fixed_ range. This is due to the release of volatile matter during carbonization, causing the decomposition of noncarbon species and the formation of abundant carbon. Some ash is removed by dissolving it in the liquid phase, which does not cause significant carbon burn-off [29]. The carbon content of the prepared carbons increased from 63 wt% to 70–80 wt%, while the hydrogen and oxygen decreased. The hydrogen content dropped from 5.3 wt% to ~2–3 wt%, and the oxygen content decreased from 23 wt% to 14–22 wt%. Sulfur was not detected in any of the samples. The altering of the elemental content of the carbon activated with these chemicals is consistent with results previously reported by Angin and Fierro et al. [30,31]. Additionally, the NPC activated by KOH had slightly higher carbon and lower oxygen content than the NPC activated by H_3_PO_4_.

The external surfaces of NPC-K and NPC-H were examined by SEM, as depicted in Figure 1a,b, respectively. Upon impregnation with KOH (Figure 1a), regular and small pores with enhanced porosity were clearly observed on the surface of the NPC-K. The activation process of the reaction between KOH and C generated H_2_, K_2_CO_3_, and a potassium compound (K or K_2_O). K_2_CO_3_ decomposes at a temperature higher than 800 °C and results in CO and CO_2_ [32]. On the contrary, NPC-H has cracks and cavities on its external surface (Figure 1b). Those cavities are caused by the evaporation of the activating reagent (H_3_PO_4_) throughout carbonization and leaving the space of previously possessed reagents. More detailed internal morphologies of NPC-K and NPC-H samples were observed by using TEM, as presented in Figure 1c,d, respectively. These depict a disordered hierarchical porous structure, which contains innumerable pores. The large white gaps between the disordered carbon layers suggest that substantial micropores and mesopores coexist in the NPC. The reaction between the carbon precursor and the activating reagent with heat treatment leads to decomposition of the volatile organic compounds, forming porous carbons. The EDX analysis provides information on the elemental surface composition, as presented in Figure 1e,f. The carbon activated by KOH revealed the presence of K on its surface (Figure 1e). The carbon activated by H_3_PO_4_ showed the presence of P on its surface (Figure 1f). In addition, small amounts of inorganic elements were detectable, such as Na, Al, Si, and Ca, which may originate from biomass precursors.

The XRD patterns of both NPC samples are illustrated in Figure 2a. The XRD patterns of both samples presented two broad peaks: a distinct peak at 24° assigned to the (002) plane and a less intense peak at 43.5° assigned to the (100) plane of the carbon material [33]. These broad peaks clearly depict the amorphous carbon structure and disorderly stack-up of carbon rings [34]. The interlayer spacing, *d*_002_, of the NPC samples was calculated by Bragg’s equation (Equation (1)):(1)$$d_{002}=\frac{\lambda }{2\sin\theta },$$
where λ is the X-ray wavelength (λ = 0.15406 nm) and θ is the peak position angle.

The crystallite height along the *c*-axis, L_c_, and the crystallite width along the *a*-axis, L_a_, were determined using the Scherrer equation (Equation (2)):(2)$$L=\frac{K\lambda }{B\;\cos\theta },$$
where L is (L_c_) or (L_a_); B is the full-width half maximum (FWHM) of the peak in radian; and K is the factor of the shape, in which K = 0.9 and 1.84 are used to calculate L_c_ and L_a_, respectively. The L_c_ and L_a_ values are based on the peaks at the planes (002) and (100), respectively.

The values of *d*_002_, L_c_, and L_a_ are shown in Table 2. The *d*_002_ value for both NPC-K and NPC-H samples was 0.37 nm, which is larger than that of graphite (*d*_002_ *=* 0.335 nm). Similar results were reported by Hadoun et al. [35] and Prahas et al. [36]. The high *d*_002_ value (*d*_002_ > 0.335 nm) is attributed to the fact that the NPC samples were disordered and had improper graphitization [37]. The L_c_ of both NPC samples was approximately 0.6 nm, corresponding to monolayer graphene and forming a graphitic platelet. In addition, the L_c_ and L_a_ values of NPC-K were smaller than those of NPC-H, which indicates that NPC-K had better porosity and thus a higher surface area than NPC-H [36].

In order to further investigate the graphitic structure of carbon, the carbon materials were further subjected to Raman spectra. As shown in Figure 2b, the Raman spectra of NPC-K and NPC-H displayed a D band and a G band at around 1350 and 1595 cm^−1^, respectively, which represent the disordered and graphitic carbon structure of the carbon materials. The D band intensity was increased by the number of sp^3^-bonded carbon atoms at the edge and the suspension of the sp^2^-bonded carbon sites [38]. The intensity of the G band depends on the number of in-plane stretching motions between pairs of sp^2^ carbon atoms. Usually, the G band is much greater than the D band; however, in noncrystalline carbons, such as NPC-K and NPC-H, the D band is mainly shown. Important data extracted from the Raman spectra in Figure 2b are listed in Table 2. The degree of disordered structure in carbon materials with a graphitic structure was obtained by the relative peak intensity ratio (R_p_) of the D to G peaks (I_D_/I_G_). Both the NPC-K and NPC-H samples exhibited an I_D_/I_G_ ratio of around 1, indicating that more amorphous carbon structures take the place of the graphitic carbon structure.

Generally, R_p_ varies inversely with L_a_ [39]. An increase in the D band intensity is correlated with a decrease in L_a_ and vice versa. The structural parameters inferred from XRD complemented the Raman results. A correlation of L_a_ values with the relative intensity of the D band was found from the Raman spectrum data as per Equation (3) [40]:(3)$$R_{p}\left(\frac{I_{D}}{I_{G}}\right)=\frac{4.4}{L_{a}}.$$

The L_a_ from Raman studies correlated well with the value of L_a_ obtained from XRD analytes using the Scherrer equation (Table 2). These results emphasize that XRD and Raman studies are similarly informative for the crystalline disorder in the carbon structure.

The N_2_ adsorption–desorption isotherms and pore size distribution of NPC-K and NPC-H and their corresponding catalysts (Pd/NPC-K and Pd/NPC-H) are shown in Figure 3. The isotherms of all samples obviously show an initial region of micropore filling, followed by multilayer physical adsorption and capillary condensation at intermediate and high relative pressures (Figure 3a). According to the IUPAC classification [41], these isotherms can be classified as a combination of type I and type IV, indicating the presence of micropores and mesopores. All isotherms exhibited a type H4 hysteresis, characteristic of slit-shaped pores. The isotherm did not show that the closure point of the hysteresis loop may be attributed to a physical phenomenon, such as irreversible uptake, owing to the presence of pores of the same width of the adsorbate molecule and/or swelling of the nonrigid porous framework of the carbon [42]. As shown in Figure 3b, most of the pores were distributed in the range of 0.7–3.0 nm, and a small distribution was also observed in the pore size range of 20–80 nm, which confirmed the presence of the predominant micropores and some mesopores in the carbon. The average D_p_ of both samples was in the range of micropores (<2 nm), 0.73 nm for NPC-K and 0.80 nm for NPC-H. Additionally, the use of KOH as an activating reagent led to an increase in the volume of both micropores and mesopores compared to the H_3_PO_4_ reagent, indicating micropore formation and micropore widening to mesopores. The S_BET_ of NPC-K was 1378 m^2^ g^−1^, which was higher than that of NPC-K (758 m^2^ g^−1^). The amorphous structure and high surface area of synthesized NPC are expected to yield smaller metal particle sizes, indicating the advantage of this material as a catalyst support for catalytic reactions [43].

FTIR spectroscopy demonstrates the existence of specific functional groups on the carbon surface. Several characteristic bands appeared in the FTIR spectrum of NPC-K and NPC-H samples, as illustrated in Figure 4. Each band was assigned to a specific functional group according to the literature [44,45,46].

The broad band located around 2900–2600 cm^−1^ is the C–H stretching vibration in the methyl and methylene groups. The presence of bands around 2300 cm^−1^, 2100 cm^−1^, and 2000 cm^−1^ can be attributed to the stretching vibration of O=C=O, C=C=O (ketene), and C=C=C (allene), respectively. The bands in the range of 1950–1600 cm^−1^ are assigned to the vibrate stretching of the C=O bond in carboxylic acids, aldehydes, ketones, lactones, and esters. A band in the 1850–1800 cm^−1^ region is characteristic of carbonyl moieties in carboxylic anhydrides. The band centered at 1545 cm^−1^ belongs to the stretched vibrations of the C=C bond in aromatic rings, a typical characteristic of carbonaceous materials. The band located at 1250–1000 cm^−1^ indicates a single C–O bond in carboxylic acid, alcohol, lactone, phenol, and ether, or C–O–C (vibration at 1055 cm^−1^ and 1028 cm^−1^) in cyclic anhydride. Due to the presence of phosphorous in the NPC-H sample, the band at 1250–1100 cm^−1^ may be attributed to the stretching mode of hydrogen-bound P=O, the stretching vibration O–C in the linkage of P–O–C, and P=OOH, and the shoulder area at 1060–1030 cm^−1^ can be defined as P^+^–O^−^ ionization linkage in the acid phosphate esters and symmetrical vibration in the P–O–P chain [47].

The wide-range XPS spectra of NPC-K and NPC-H are depicted in Appendix A. O 1s and C 1s peaks were observed in the XPS profiles of both NPC samples. In addition, a characteristic peak of P 2p was visible at 133 eV in the XPS profile of the NPC-H sample, indicating the presence of P residues on its surface. The XPS spectra of C 1s presented a complicated envelope that identified several carbon species at the carbon surface. Figure 5a,b present the deconvolution of C 1s spectra into six components used in NPC-K and NPC-H analysis. The components represent carbon carbides; graphitic carbon (C–C and C–H, which are indistinguishable); C–O link in ether, alcohol groups and C–O–P linkage; carbon species in carbonyl groups (C=O); carboxylic and ester functional groups (O–C=O); and the satellite shaking due to π–π* shifts in the aromatic rings [46,48]. The peak BE of these chemical states has been well documented in the literature [46,48,49]. The peak BE may slightly shift, compared to what is reported in the literature, due to the chemical nature of the neighboring atoms on an individual surface. The BE and atomic percentage of each carbon state composing the NPC surfaces are listed in Table 3.

A broad O 1s peak indicates the presence of several chemical states of oxygen (Figure 5c,d). The O 1s peak can be deconvoluted into four components (Table 3). The peak A located in the range of 530.2 eV showed the presence of OH^−^ anion species on the carbon surface. The peak B at BE around 531.4–531.7 eV is ascribed to oxygen double bonded to carbon (C=O) and unbound oxygen in the phosphate group (P=O). The peak C of BE around 533.1–533.2 eV can be assigned to oxygen singly bonded to carbon (–O–) in phenols and ethers and/or in C–O–P groups. The peak D at BE = 536.2 eV is attributed to chemisorbed water and/or occluded CO and CO_2_ [49,50]. It can be seen that the percentage of oxygen atomic concentration on the NPC-H surface is much higher than that of NPC-K, indicating a greater oxidizing surface on NPC-H. The OH^−^ anion, which was the main oxygen species, was present only on the NPC-H surface (Peak A). This was attributed to the greater electronegativity of oxygen compared to the C atom, which leads to electron transfer from the neighboring carbon atom to the oxygen atom, making the carbon atom itself positive in NPC-H. Similar percentages of double-bonded oxygen (Peak B) and single-bonded oxygen (Peak C) were present in both synthesized carbons.

P 2p signals at BE of ~133 eV can be observed only in the NPC-H sample. The P 2p peak was deconvoluted into four components, as shown in Figure 5e, and the details are shown in Table 3. The main component with BE = 133.0 eV (Peak E) is assigned to pentavalent tetra-coordinated phosphorus (PO_4_ tetrahedra), which is present in phosphate and pyrophosphate [50]. C–P bonding in phosphonate compounds is also seen in this region (Peaks C and D) [51]. Peak F with BE = 133.8 eV can be assigned to metaphosphates [50]. These results clearly confirm that the candidates for such phosphorus-containing compounds are polyphosphate compounds. P may also play an electron-donor role, due to having the smallest electronegativity compared to the neighboring atoms (C and O), resulting in electron deficiency. From FTIR and XPS studies, we can infer that the presence of oxygen- and phosphorus-containing groups on the carbon surface may affect the acidity of the catalyst, resulting in the good catalytic performance.

The PZC and acidity of the prepared NPC materials were also determined, and the corresponding values are presented in Table 4. NPC-K and NPC-H possess a clearly acidic character, with a PZC below 7. According to [52], the PZC value is inversely proportional to the total oxygen content of carbon. The bulk oxygen (Table 1) and surface oxygen (Table 3) content of the NPC-H were higher than those of the NPC-K. As a result, NPC-K had a weakly acidic character, with low acidity (0.17 mmol g^−1^) and a PZC value of 4.8. In contrast, NPC-H, due to the presence of phosphorus-containing compounds and a much higher oxygen content, is a strongly acidic carbon material, as revealed by its high acidity (0.84 mmol g^−1^) and low PZC value of 2.6. The carbon surface may reveal negative and positive charges depending on the pH of the surroundings, becoming an important variable for regulating the metal–support interaction [23,53].

### 3.2. Properties of Pd/NPC Catalyst

The Pd loading of the reduced Pd/NPC-K and Pd/NPC-H catalysts was 4.84 and 4.92 wt%, respectively, which was close to the nominal value (5 wt%) (Table 4). Figure 6 depicts the XRD patterns of the reduced Pd/NPC-K and Pd/NPC-H catalysts. The characteristic peaks of amorphous carbon located at 2θ of 24.5° and 43.6° are attributed to the NPC support. The three peaks were observed at 2θ of 40°, 46.5°, and 68°, corresponding to the face-centered cubic Pd in the planes of (111), (200), and (220), respectively (PDF# 46-1043). The peak intensity of metallic Pd in the Pd/NPC-K catalyst was obviously higher than that of the Pd/NPC-H catalyst, indicating a larger crystalline size of Pd in Pd/NPC-K catalyst.

The TEM images of the reduced Pd/NPC-K catalyst showed coexisting large Pd particles in the 10–13 nm size range. The smaller Pd particles in a size range of 6–8 nm were observed in the reduced Pd/NPC-H catalyst (Figure 7a,b, respectively). The Pd crystalline size from the XRD results is in accordance with the Pd particle size in the TEM image of both catalysts, as seen in Table 4. Dissolving the Pd(NO_3_)_2_ precursor in deionized water resulted in an aqueous solution of [Pd(H_2_O)_4_]^2+^ cations with a pH range of 1.57–1.60. On Pd impregnation, cation exchange between the [Pd(H_2_O)_4_]^2+^ species and positive charge of the NPC surface took place under acidic conditions. The large Pd particle and low Pd dispersion of the Pd/NPC-K catalyst could be ascribed to the weak interactions between the Pd precursor and the NPC support. By contrast, the strong interactivity between the Pd cation and carbon may retard the agglomeration of Pd particles, resulting in the small Pd particles of the Pd/NPC-H catalyst.

The XPS technique was used to analyze the chemical states of Pd in the reduced Pd/NPC catalysts, which are presented in Figure 8. The 3d XPS spectra of Pd revealed predominant doublet peaks at around 336 and 341 eV, attributed to Pd^0^ 3d_5/2_ and Pd^0^ 3d_3/2_, respectively. Two minor peaks observed at around 337 and 343 eV are assigned to Pd^2+^ species 3d_5/2_ and Pd^2+^ species 3d_3/2_, indicating that the partial Pd nanoparticles on the Pd/NPC catalyst surface were oxidized by exposure to air during the XPS measurement [54,55]. The Pd^0^ 3d_5/2_ of the Pd/NPC-K and Pd/NPC-H catalyst, located at 335.6 eV and 335.7 eV, respectively, positively shifted by 0.6–0.7 eV compared to those of the bulk Pd metal (BE = 335.0 eV), indicating electron transfer from Pd particles to neighboring atoms, increasing the acidity of Pd [56]. The relative distribution of the surface Pd species (Pd^0^/Pd^2+^) ratios of the reduced catalyst is summarized in Table 4. The Pd^0^/Pd^2+^ ratio of the Pd/NPC-K catalyst was 2.2, which is similar to that of the Pd/NPC-H catalyst (Pd^0^/Pd^2+^ = 2.1). The XPS signals of Pd 3d in the Pd/NPC-H catalyst were less intense than those of the Pd/NPC-K catalyst because the small Pd particles on the Pd/NPC-H catalyst could be covered by the carbon frame compared with the large Pd particles on the Pd/NPC-K catalyst.

The acidities of the Pd/NPC-K and the Pd/NPC-H catalysts were 0.32 and 1.93 mmol g^−1^, respectively, which were significantly higher than those of the NPC supports (Table 4). This was attributed to the presence of electron-deficient Pd and more oxygenated species forming on the carbon surface during the impregnation [57]. The wide-range XPS spectra of both catalysts obviously showed the increase in the oxygen surface content as depicted in Appendix A. Additionally, the S_BET_, V_total_, V_micro_, and V_meso_ of the NPC-K supports were about twice those of the NPC-H supports (Table 4). After Pd impregnation, the S_BET_, V_total_, and D_p_ values of the corresponding Pd catalysts were significantly lower than those of the NPC support due to the coverage of the surface and pore mounts of NPC by Pd particles. However, the decrease of the S_BET_ of both carbons in the corresponding Pd catalyst was similar, about 29%; thus, it can be implied that the textural properties had a lesser effect on the Pd dispersion. These results suggested that the size of the Pd particle of these Pd/NPC catalysts is predominantly influenced by the surface chemistry of the NPC support. In addition, the S_BET_ of these catalysts remained high, in the range of 500–900 m^2^ g^−1^. The Pd particle size (6–13 nm) of both Pd catalysts was obviously larger than the pore size of the prepared NPC supports (0.7–0.8 nm), suggesting that most Pd particles occupied the NPC surface; thus, the reaction also took place on the catalyst surface. It is thus inferred that the pore size of NPC supports may not affect the activity and product selectivity of poly-FAME partial hydrogenation [25,58].

### 3.3. Partial Hydrogenation of Poly-FAME

The partial hydrogenation of poly-FAME in palm-BDF, to yield H-FAME, was performed on the Pd/NPC-K and Pd/NPC-H catalysts to evaluate the hydrotreating performance of these catalysts. Methyl linoleate (C_18:2_; 8.5 wt%) and methyl linolenate (C_18:3_; 0.1 wt%) were the major poly-FAME in palm-BDF, causing low stability of oxidation. The *cis*-methyl oleate (*cis*-C_18:1_; 36 wt%) and the *trans*-methyl oleate (*trans*-C_18:1_; 0.2 wt%) were the main monounsaturated FAME in palm-BDF. The main saturated FAME in palm-BDF was methyl palmitate (C_16:0_) and methyl stearate (C_18:0_), with amounts of 45.5 wt% and 4.8 wt%, respectively. A previous study reported relative autoxidation rates of linolenate (C_18:3_), linoleate (C_18:2_), and oleates (C_18:1_) of 98, 41, and 1, respectively [59]. In addition, the melting point of the corresponding C18 FAME components increased in the order of *cis*-C_18:1_ (–20.2 °C), *trans*-C_18:1_ (~10 °C), and C_18:0_ (~38 °C), which correlates with the cold flow properties of palm-BDF. Therefore, the partial hydrogenation of poly-FAME (including C_18:3_ and C_18:2_) to produce *cis*-C_18:1_ and diminish *trans*-C_18:1_ and C_18:0_ formation was favored to enhance the oxidation stability and preserve the cold flow properties of biodiesel.

Figure 9 presents the poly-FAME conversion with reaction time of the Pd/NPC-K and Pd/NPC-H catalysts. The Pd/NPC-H catalyst could achieve 90% poly-FAME conversion at 45 min, whereas the Pd/NPC-K catalyst required a longer time to reach the same conversion rate of poly-FAME components. The initial rates can be calculated by dividing the molar amount of consumed C_18:2_ components by the Pd mass in the first 10 min of the reaction. The initial rate of the Pd/NPC-H catalyst (645 mmol g_pd_^−1^ s^−1^) was dramatically higher than that of Pd/NPC-K (279 mmol g_pd_^−1^ s^−1^) (Table 5), indicating that the Pd/NPC-H catalyst had the highest hydrogenation activity of poly-FAME components.

The hydrogenation activity of poly-FAME on the Pd/NPC-H and the Pd/NPC-K catalysts varied according to the Pd particle size and acidity of the catalyst. The Pd/NPC-H catalyst was superior to the Pd/NPC-K catalyst in terms of the hydrogenation activity of poly-FAME because the small Pd particles and high acidity of Pd/NPC-H make the hydrogen and poly-FAME molecules highly accessible. Additionally, the electron-rich double bonds in poly-FAME molecules favor adsorbing on the acidic surface of the catalyst, resulting in higher catalytic activity for Pd/NPC-H. This behavior has been reported in the literature [60,61,62,63].

**Table 5 nanomaterials-11-01431-t005:** Initial rate, FAME composition, and biodiesel properties of palm-BDF and H-FAME over the prepared Pd/NPC catalysts at a poly-FAME conversion level of 90%.

		Pd/NPC-K ^1^	Pd/NPC-H ^1^	2% Pd/CA ^2^	2% Pd/AC ^3^
Particle size (nm)		11	7	10	17
Initial rate (mmol g_pd_^−1^ s^−1^) ^4^		279	645	-	-
C_18:1_ selectivity (%)		82	84	-	-
Reaction time (min)		100	45	90	90
FAME composition (wt%)	palm-BDF	H-FAME			
C_18:3_	0.1	0	0	-	0
C_18:2_	8.5	0.9	0.9	-	0.3
*total* C_18:1_	36.2	39.9	41.7	-	31.6
*cis*-C_18:1_	36.1	30.6	30.1	-	14.3
*trans*-C_18:1_	0.1	9.3	11.6	-	17.4
C_18:0_	4.8	8.9	7.9	-	-
Poly-FAME	8.6	0.9	0.9	0	0.3
Mono-FAME	36.2	39.9	41.7	4.7	31.6
Sat-FAME	52.6	58.8	57.6	95.7	66.0
Biodiesel properties					
Oxidation stability (h)	13	65	65	41	33
Cloud point (°C)	14	17	16	26	23

^1^ The reaction conditions were temperature: 80 °C, H_2_ pressure: 0.5 MPa, catalyst weight: 0.2 g, and oil weight: 200 g (corresponding to a catalyst weight of 0.5 wt% in the reaction). ^2,3^ The reaction conditions were: temperature: 120 °C, H_2_ pressure: 0.4 MPa, catalyst weight: 1.5 g, and oil weight: 100 g (corresponding to a catalyst weight of 3 wt% in the reaction). ^2^ The 2% Pd/CA catalyst represented the carbon aerogel (CA)-supported Pd catalyst with a Pd loading of 2 wt% [5]. ^3^ The 2% Pd/AC catalyst represented the activated carbon (AC)-supported Pd catalyst with a Pd loading of 2 wt% [64]. ^4^ Calculated after 10 min of reaction time.

Figure 10 shows the amounts of poly-FAME, *total*-C_18:1_ (total amount of *cis*-C_18:1_ and *trans*-C_18:1_), *cis*-C_18:1_, *trans*-C_18:1_, and C_18:0_ at various poly-FAME conversion levels. The C_18_ FAME compositions obtained by H-FAME on the Pd/NPC-K and Pd/NPC-H catalysts revealed a similar alteration in the same poly-FAME conversion levels. The poly-FAME contents gradually decreased with a substantial increase in the *cis*-C1_8:1_, *trans*-C_18:1_, and C_18:0_ contents. The highest *cis*-C1_8:1_ level was produced at 40% of poly-FAME conversion and then continuously decreased after that, leading to a rapid increase in the *trans*-C_18:1_ and C_18:0_ contents. On the other hand, the *total*-C_18:1_ continuously increased to the maximum level at 90% of poly-FAME conversion; the *total*-C_18:1_ decreased significantly, along with a sharp increase in C_18:0_. The consecutive reaction behaviors from C_18:3_ to C_18:2_, C_18:2_ to C_18:1_, and then C_18:1_ to C_18:0_ are consistent with the literature [64,65]. Therefore, a 90% poly-FAME conversion was appropriate as a compromise between the lowest poly-FAME content (0.9 wt%), the highest content of *total*-C_18:1_ (40–42 wt%), and the acceptable content of C_18:0_ (<10 wt%).

The selectivity toward the C_18:1_ component was calculated using Equation (4), which revealed 82% for Pd/NPC-K and 84% for Pd/NPC-H at 90% poly-FAME conversion (Table 5). This suggested that the product selectivity of the Pd/NPC-K and Pd/NPC-H catalysts was similar.
(4)$${\mathrm{Selectivity}\;\mathrm{to}\;C}_{18:1}\left(\%\right)=\left(\frac{{\mathrm{Total}\;\mathrm{amount}\;\mathrm{of}\;C}_{18:1}}{{\mathrm{Total}\;\mathrm{amount}\;\mathrm{of}\;C}_{18:1}{\;\mathrm{and}\;C}_{18:0}}\right)\times 100$$

The use of relatively low-structured, carbon-supported Pd catalysts such as carbon aerogels (CA) and activated carbon (AC) has been reported for the partial hydrogenation of poly-FAME [5,64]. Although the reaction was performed at a high catalyst concentration and reaction temperature, 2% Pd/CA and 2% Pd/AC with larger Pd particle size (10–17 nm) had lower hydrogenation activity than the Pd/NPC-H catalyst but similar hydrogenation activity as the Pd/NPC-K catalyst. Both the 2% Pd/CA and 2% Pd/AC catalysts presented high selectivity toward sat-FAME, which are undesired products for upgrading the biodiesel quality. It can be said that the Pd/NPC-H catalyst with ~7 nm Pd particle size is a highly active and selective catalyst for the partial hydrogenation of poly-FAME into mono-FAME under the studied reaction conditions.

The FAME compositions in the produced H-FAME at 90% poly-FAME conversion are given in Table 5, which correlates their oxidation stability and cloud point. It is noted that the oxidation stability of general palm-BDF is <5 h; however, its oxidation stability increased to 13 h by the addition of antioxidants (Table 5), making the biodiesel commercial grade [64,65]. The oxidation stability of palm-BDF already meets the existing biodiesel standards, including the international standard EN12,214 (>8 h), ASTM D6751 (>3 h), and the Thai national standard (>10 h) [66,67]. Enhancing the oxidation stability of palm-BDF for use in highly blended fuels such as B10–B30, is necessary to support the Alternative Energy Development Plan (AEDP, 2018) of the Thai government [68]. The H-FAME obtained with all the catalysts showed a high oxidation stability of 65 h with an admissible cloud point, representing the cold flow properties. The cloud point of H-FAME was 16–17 °C, close to the specified 16 °C in the Thai national standard. These properties were due to most of the poly-FAME components being eliminated to be the corresponding C_18:1_ component, also minimizing the deep hydrogenation. Additionally, the resulting H-FAME cloud point could be tuned below 16 °C by further distillation or precipitation to remove the saturated components [69,70].

## 4. Conclusions

The NPC-K and NPC-H supports were prepared from the CTF biomass via hydrothermal carbonization-assisted chemical activation using KOH and H_3_PO_4_, respectively. Both NPC materials had relatively low-structured carbon with micropores, whereas NPC-H presented a superior oxygen concentration and phosphate functionalities on the material’s surface. Different types of NPC support provided different catalytic properties, particularly metal particle size and acidity, and, as a result, different catalytic activity was observed. The Pd/NPC-H catalyst with small Pd particles (7 nm) and high acidity presented high hydrogenation activity because of the high accessibility of the polyunsaturated FAME molecules on Pd active sites. By contrast, the larger Pd particles and lower acidity of Pd/NPC-K presented lower activity for this reaction. In terms of the desired product (C_18:1_), the highest content could be found at 90% of polyunsaturated FAME conversion over all the catalysts, resulting in significantly increased oxidative stability. Therefore, it was clearly demonstrated that low-cost CTF-derived materials with microporosity were potential catalytic supports for the partial hydrogenation of polyunsaturated FAME.

## Figures and Tables

**Figure 1 nanomaterials-11-01431-f001:**
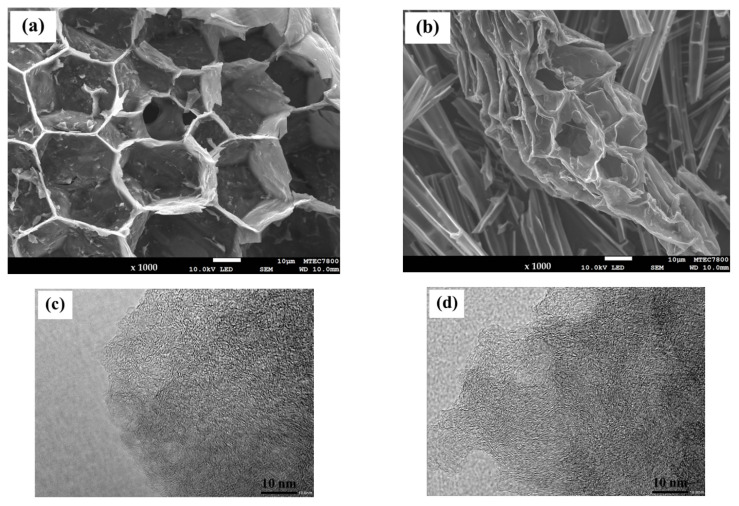
SEM and TEM images, and EDX spectra of (**a**,**c**,**e**) NPC-K and (**b**,**d**,**f**) NPC-H.

**Figure 2 nanomaterials-11-01431-f002:**
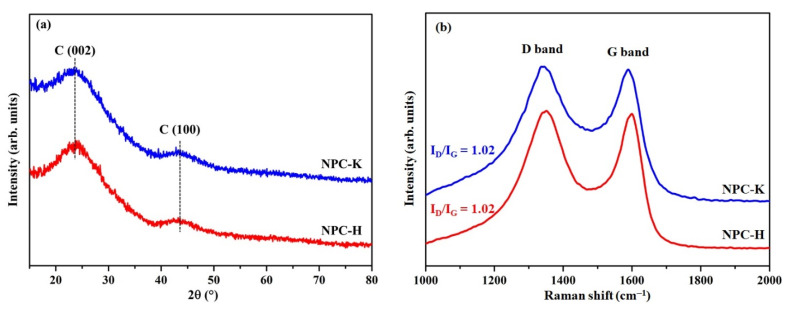
(**a**) XRD pattern and (**b**) Raman spectra with calculated ID/IG ratios of the NPC-K (blue curves) and NPC-H (red curves) samples.

**Figure 3 nanomaterials-11-01431-f003:**
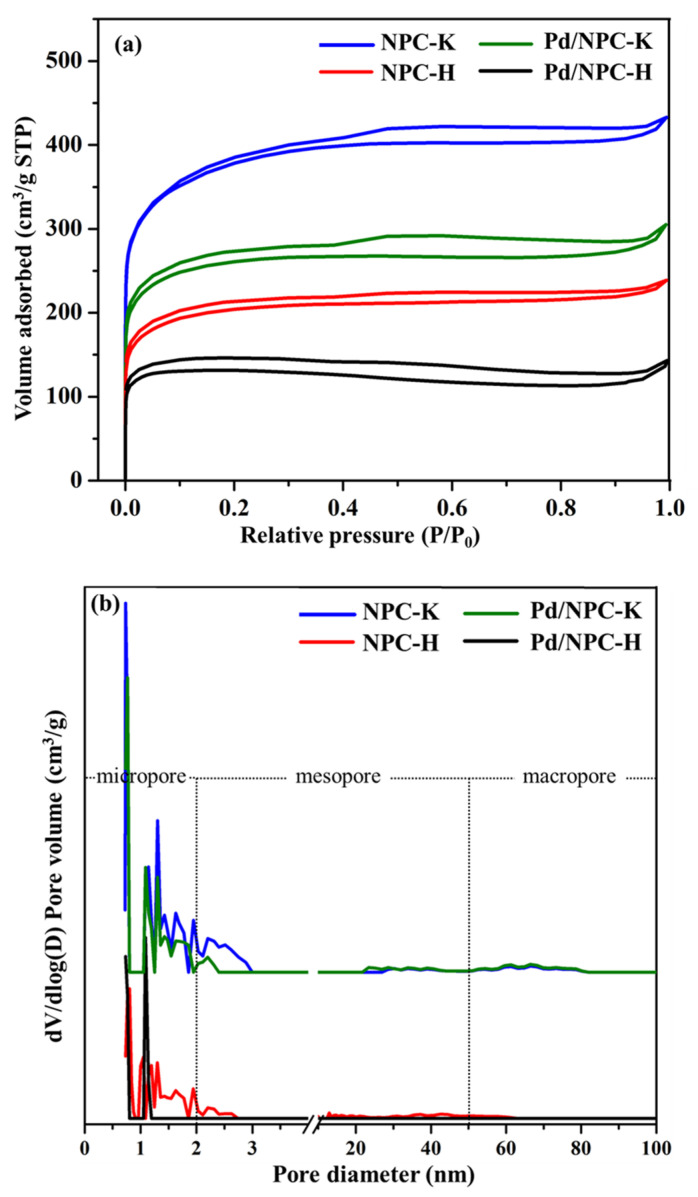
(**a**) N_2_ adsorption–desorption isotherm and (**b**) pore size distribution plot of NPC-K, NPC-H, Pd/NPC-K, and Pd/NPC-H.

**Figure 4 nanomaterials-11-01431-f004:**
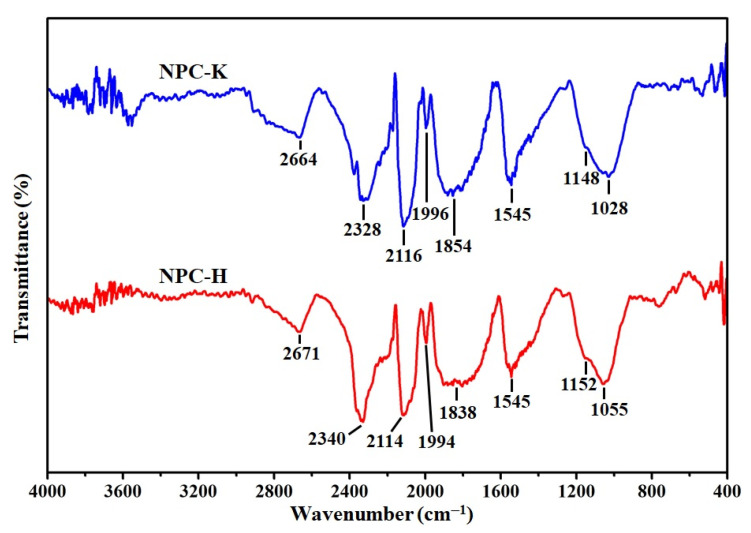
FTIR spectra of NPC-K and NPC-H samples.

**Figure 5 nanomaterials-11-01431-f005:**
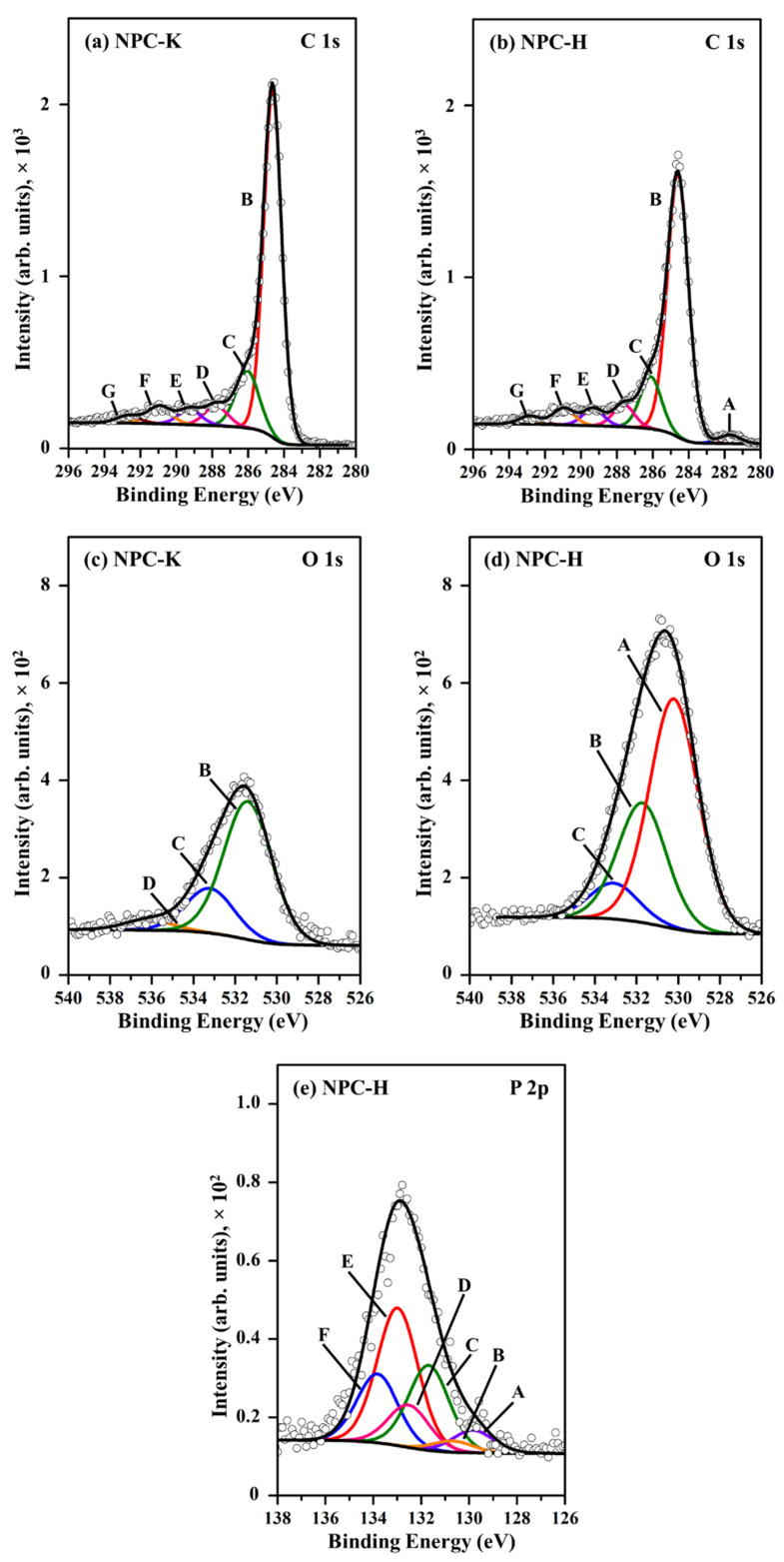
XPS spectrum of C 1s (**a**,**b**), O 1s (**c**,**d**), and P 2p (**e**) peak of NPC-K and NPC-H samples, see Table 3.

**Figure 6 nanomaterials-11-01431-f006:**
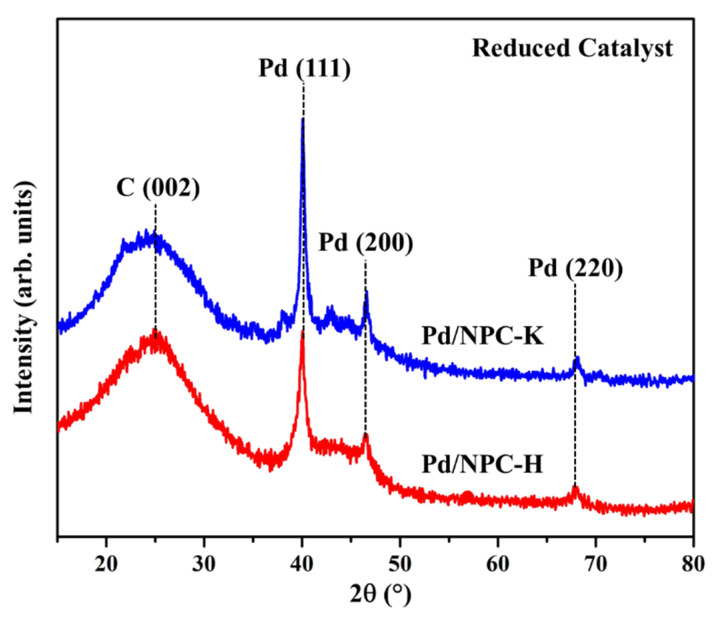
XRD patterns of the reduced Pd/NPC-K (blue curves) and Pd/NPC-H (red curves) catalysts.

**Figure 7 nanomaterials-11-01431-f007:**
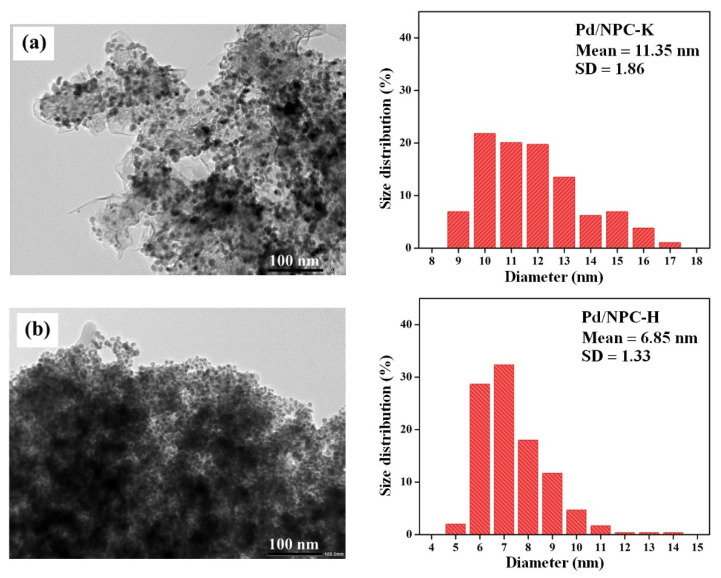
TEM micrographs and Pd size distribution of the (**a**) Pd/NPC-K and (**b**) Pd/NPC-H catalysts.

**Figure 8 nanomaterials-11-01431-f008:**
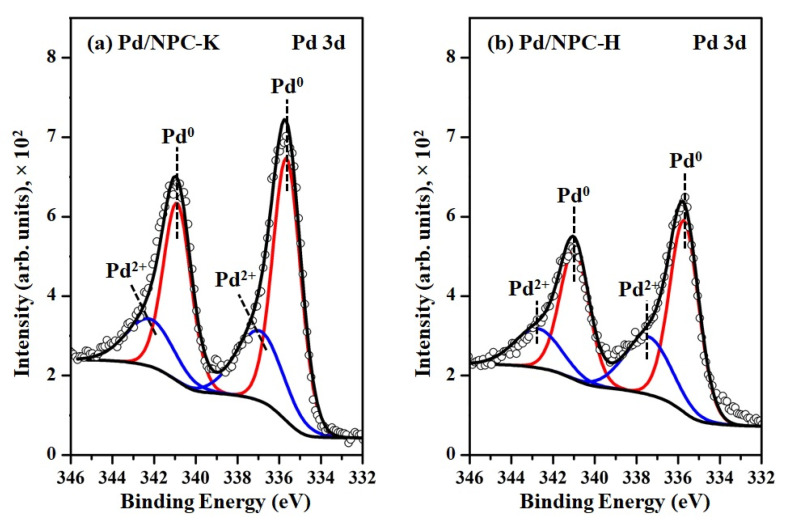
XPS spectra of Pd 3d in the reduced (**a**) Pd/NPC-K and (**b**) Pd/NPC-H catalysts.

**Figure 9 nanomaterials-11-01431-f009:**
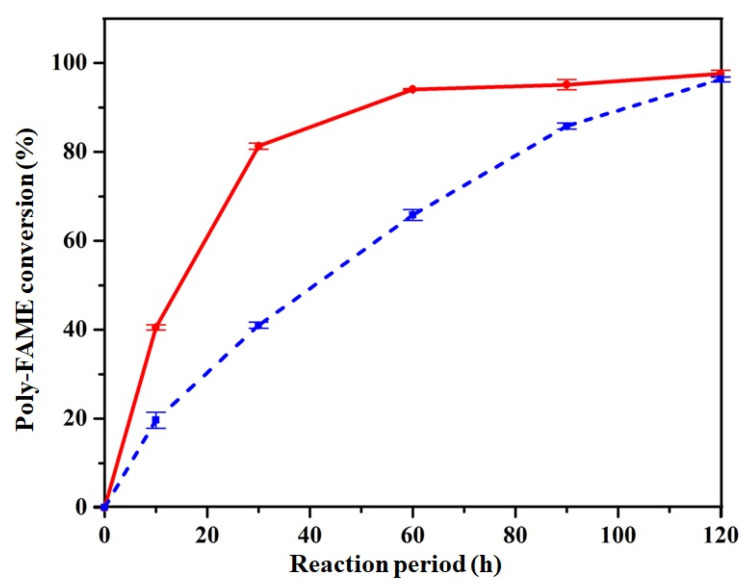
Poly-FAME conversion at various reaction times of Pd/NPC-H (solid line) and Pd/NPC-K (dashed line) catalysts.

**Figure 10 nanomaterials-11-01431-f010:**
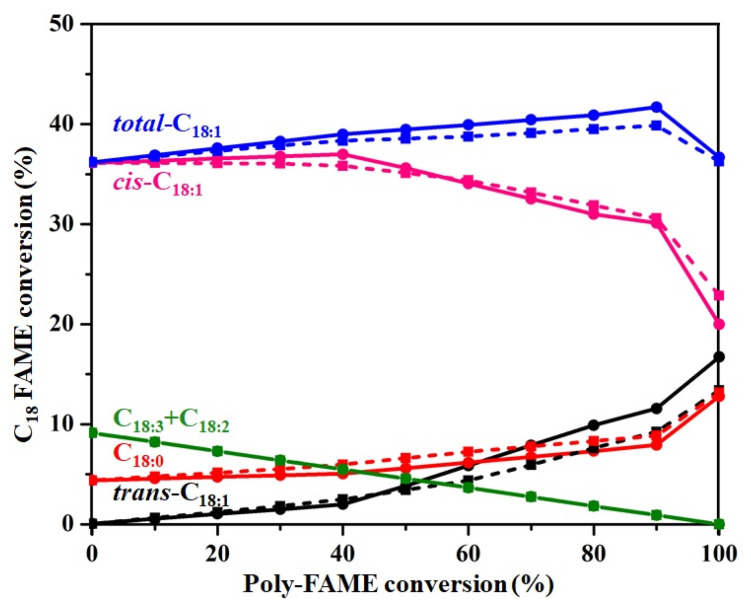
C18 FAME composition in the H-FAME product at different poly-FAME conversion levels over the Pd/NPC-H (solid line) and Pd/NPC-K (dashed line) catalysts.

**Table 1 nanomaterials-11-01431-t001:** Proximate and elemental analyses of the CTF precursor, NPC-K, and NPC-H.

Sample	Proximate Analysis (wt%)					Elemental Analysis ^1^ (wt%)			
	Moisture	Volatiles	C_fixed_	Ash	C	H	N	S	O ^2^
Raw CTF	7.53	75.5	12.1	4.87	54.2	6.56	1.88	0.24	32.3
Hydrochar	5.72	58.5	29.5	6.25	63.4	5.25	1.75	0.20	23.2
NPC-K	4.42	20.5	73.0	2.12	79.7	2.50	1.72	ND ^3^	14.0
NPC-H	3.65	24.7	69.2	2.45	72.2	2.26	1.63	ND	21.5

^1^ on dry basis. ^2^ O% = 100—(C% + H% + N% + S% + Ash%). ^3^ Not detected.

**Table 2 nanomaterials-11-01431-t002:** Structural parameters of NPC-K and NPC-H, deduced from XRD diffraction and Raman spectra.

Sample	*d*_002_ (nm)	L_c_ (nm)	L_a_ ^1^ (nm)	R_p_ (I_D_/I_G_) ^2^	L_a_ (4.4/R) ^3^ (nm)
NPC-K	0.37	0.58	2.87	1.02	4.31
NPC-H	0.37	0.62	3.15	1.02	4.31

^1^ L_a_, Crystalline size calculated from XRD. ^2^ R_p_, Ratio of peak intensity from Raman spectra (I_D_/I_G_). ^3^ L_a_, Crystalline size calculated from Raman spectra.

**Table 3 nanomaterials-11-01431-t003:** Deconvolution of XPS spectra of NPC-K and NPC-H samples.

Region	Peak	NPC-K		NPC-H		Assignment
		BE (eV)	wt%	BE (eV)	wt%	
C 1s	A	-	-	281.7	1.75	Carbide
	B	284.6	62.9	284.6	53.0	Graphitic carbon
	C	286.1	13.3	286.1	10.5	C–O– in phenol and alcohol
	D	287.8	4.64	287.6	4.48	C–O– in ether, C-O-P
	E	289.3	3.50	289.3	3.46	O–C=O in carboxyl or ester
	F	291.0	4.04	291.0	3.37	C=O/C=C in carbonate, occluded CO
	G	292.7	1.75	292.9	1.66	π–π* transition due to conjugation
O 1s	A	-	-	530.2	11.3	OH^−^ anion
	B	531.4	7.04	531.7	5.87	C=O in carbonate, P=O
	C	533.2	2.28	533.1	1.77	C–O– in aromatic rings, phenols and ethers, C–O–P
	D	536.2	0.53	-	-	Chemisorbed water
P 2p	A (2p_3/2_)	-	-	129.8	0.17	P
	B (2p_1/2_)	-	-	130.7	0.09	P
	C (2p_3/2_)	-	-	131.7	0.66	C–P bonding
	D (2p_1/2_)	-	-	132.5	0.33	C–P bonding
	E (2p_3/2_)	-	-	133.0	1.07	Phosphates and pyrophosphates
	F (2p_1/2_)	-	-	133.8	0.53	Metaphosphates

**Table 4 nanomaterials-11-01431-t004:** Characteristics of the NPC-K and NPC-H materials and the Pd/NPC-K and Pd/NPC-H catalysts.

Sample	PZC	Acidity ^1^ (mmol g^−1^)	Pd Loading ^2^ (wt%)	Pd Particle Size (nm)		Pd^0^/Pd^2+^ Ratio ^3^	S_BET_ (m^2^ g^−1^)	D_p_ (nm)	V_total_ (cm^3^ g^−1^)	V_micro_ (cm^3^ g^−1^)	V_meso_ (cm^3^ g^−1^)
				XRD	TEM						
NPC-K	4.8	0.17	-	-	-	-	1378	0.73	0.67	0.51	0.16
NPC-H	2.6	0.84	-	-	-	-	758	0.80	0.37	0.29	0.08
Pd/NPC-K	-	0.32	4.84	11.2	11.4 ± 1.9	2.2	979	0.64	0.43	0.37	0.05
Pd/NPC-H	-	1.93	4.92	6.82	6.9 ± 1.3	2.1	535	0.73	0.19	0.17	0.00

^1^ Measured using NH_3_-TPD. ^2^ Determined using ICP. ^3^ Determined using XPS.

## Data Availability

Data is contained within the article or Appendix A.

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
