# Peer review of "Biomass Nanoporous Carbon-Supported Pd Catalysts for Partial Hydrogenation of Biodiesel: Effects of Surface Chemistry on Pd Particle Size and Catalytic Performance"

_nanomaterials, 2021, doi:10.3390/nano11061431_

Round 1
Reviewer 1 Report
The ms reports a study regarding nanoporous Carbon Supported Pd catalysts for partial Hydrogenation of Biodiesel. For instance the ms dealt with the effects of Surface Chemistry of Pd NPs and Catalytic Performance.
I struggle to understand the conclusions made by the authors and specifically how they conclude based on the data provided here that "These results suggested that the size of the Pd particle of these Pd/NPC catalysts is predominantly influenced by the surface chemistry of NPC support, affecting the PZC value, while the textural properties inferior influence". Also, The English is very bad in many sections of the ms, this makes the reading extremely difficult. The ms contains so many data and a good portion of these data are not so relevant, hence they can be moved to a supporting information section. Overall, this ms needs rewriting and to be restructured in a better fashion.
Reviewer 2 Report
The manuscript by Udomsapet al. describes the preparation of carbonaceous supports from cattail flowers and their use for deposition of Pd particles. The obtained catalysts were further tested for partial hydrogenation of biodiesel. The paper is quite interesting, however, the authors should clearly highlight the novelty of their research over other reported works. The number of used research techniques is a big advantage of this work, but the paper shows also several weak points. In addition, the authors characterized the obtained supporting materials carefully, but did not sufficiently correlate the obtained physicochemical results of carbons with the Pd deposition data. In general, in my opinion, the paper is publishable in Nanomaterials, however, only after major revision.
Some of my comments are given below (according to the manuscript layout and not the importance of the comments):
- 33: they are inert in both acid and base media, delaying the active phase sintering and simplified recovering – I am not sure if there is a clear cause-effect relationship between these things (I mean inertness of carbons and reduced active phase sintering or ease of catalyst recovering).
- 54: “This method….” – it is not clear if the authors mean physical or chemical activation.
- 54-56: According to the authors, physical activation is responsible for developing microporosity, and chemical activation develops mesoporosity This is not true. In addition, microporosity/mesoporosity can depend on an activating agent used in chemical activation (e.g., Appl. Catal., A 452, 2013, 39-47, or the reference no. 13). In general, the introduction to carbon production methods is superficial, it should be deepen.
- In my opinion, the introduction part is a bit chaotic. This part must be improved.
- Please, stress novelty of the research.
- 116: More details of hydrothermal carbonization of cattail flowers should be given, e.g., it is not clear if the obtained hydrochar was filtered and rinsed after the process.
- Is the hydrothermal step necessary for the carbon preparation procedure? What is its role? To investigate this the authors should compare the materials obtained using 2 approaches – with hydrothermal step and by impregnation of cattail flowers directly.
- 125: Please, explain abbreviations when they are used for the first time, e.g., NPCs, BDF.
- English needs to be improved (e.g., “the prepared NPC dried at 110 °C before…”, “The low ash content is desired in carbon which is represents amount of inorganic material from the biomass precursor”).
- 131: Palladium (II) nitrate should be palladium(II) nitrate
- 132: the carbon supports were impregnated with a solution of palladium(II) nitrate.
- 106: the authors use CF as an abbreviation of cattail flower. Maybe, they should consider the use of another symbol, as CF is typically applied for carbon fibers.
- I have always problems with comparing SEM images demonstrating the different morphology. Taking another spot from the sample other images can be shown. In how far the shown images are representative? Other magnifications, e.g., 20 or 50 μm)?
- How the authors can distinguish slit-shape pores in the TEM figures?
- The possibility of measuring the Pd particle size from TEM is highly doubtful.
- Over hydrogenation – overhydrogenation?
- Was the O content measured or calculated – this should be specified. What about the content of S?
- The obtained values of O content (over 16 and 23%) are a bit surprising taking into account the conditions of carbon preparation, i.e., the temperature of synthesis (at the temperature of 900 oC, most of oxygen groups is decomposed). Could the authors explain this phenomenon? Moreover, the char obtained after hydrothermal carbonization should be investigated, as, in fact, this is a real precursor of NCPs.
- What was the yield of the prepared carbonaceous materials (both in hydrothermal process and in the thermal treatment process conducted at 900 oC)?
- The analysis of the isotherms presented in Fig. 3a proves that the hysteresis loops were not closed. Explain.
- 353: The authors wrote that both samples (NPC-K and NPC-H) contain P. What is the source of P in NPC-K? This is simply an error.
- The authors characterize in details the obtained carbonaceous supports. However, they do not correlate these results with the data on Pd deposition. Meanwhile, it is know that the presence of different types of acidic groups on carbons may influence the dispersion of metals supported on carbons. This part should be improved.
Reviewer 3 Report
Comments:
This manuscript reported by Riva et al demonstrates the preparation and characterization of two types of Cattail flower-derived nanoporous carbon (NPC) activated with KOH and H3PO4, termed as NPC-K and NPC-H, respectively. Both NPC-K and NPC-H are treated as catalyst support to load Pd particles to be corresponding Pd/NPC catalysts, which gave catalytic activity for the polyunsaturated fatty acid methyl ester (poly-FAME) hydrogenation. The material properties and catalytic performances of Pd/NPC-K and Pd/NPC-H catalysts are discussed well. However, lack of novelty makes this work unsuitable for publication in Nanomaterials. Further, there are also questionable points that need to be concerned.
- In the IR spectra, the authors claimed that the bands at 1250-1100 cm^-1 and 1060-1030 cm^-1 may be attributed to the vibrations of phosphate-derivatives, such as acid phosphate esters. However, it is noted that the IR spectra of NPC-K and NPC-H are almost identical. How it is possible for NPC-K? It does not make sense.
- The FT-IR and XPS spectra inferred the presence of phosphate-derivatives as the components on the carbon surface of NPC-H. However, the elementary analysis showed 100 wt.% of the sum of C, H, N, and O in NPC-H. How about the wt.% of P in NPC-H?
- Full XPS spectra of four materials should be provided.
- Elementary analysis of Pd/NPC-K and Pd/NPC-H catalysts should be provided.
- The results of partial hydrogenation of poly-FAME performed over Pd/NPC-K and Pd/NPC-H should be compared with those in the literature reports.
Round 2
Reviewer 1 Report
The authors performed a thorough revision of the ms which I think now meets the criteria for publication in the journal. However, I do believe some of the info presented in the main text could be moved into the SI file (just a personal preference).
Author Response
Reviewer 1 report : The authors performed a thorough revision of the ms which I think now meets the criteria for publication in the journal. However, I do believe some of the info presented in the main text could be moved into the SI file (just a personal preference).
Author response: We are grateful for your acceptance. We have realized for your suggestion. However, we have decided to maintain all the results presented in the main text because they are critical properties of activated carbon that should be taken into account before using it in different applications. These results will allow readers to shift the paradigm for future investigations into the development of the nanoporous carbon supports to produce the supported metal catalyst for various chemical reactions.
Thank you for your time and valuable feedback.
Best regards,
Parncheewa Udomsap
Reviewer 2 Report
I would like to thank the authors for their efforts in improving the manuscript. In my opinion, the paper looks significantly better. However, some small errors can still be find in the manuscript. Before the publication process, please check again the English style and the text consistency. My suggestions:
(17) The surface chemistry properties of NPC were the main factors influencing the Pd particle size
(19) Pd particles
(19) The use of Pd nanoparticles supported on the rich-functionalized surface carbons obtained by H3PO4 activation led to superior catalytic activity for the polyunsaturated fatty acid methyl ester hydrogenation
(104) According to the physical properties of CTF biomass, which is a short fiber and low bulk density substance,
(111) however, the chemical process is preferable due to a significant development of the surface area and the higher carbon yield (give a reference).
(113) can be obtained (not adjusted)
(121) surface areas, ………….yields
(127) the textural properties of …………….
(131) activating the sample with KOH at the temperature of 900 °C
(136) ZnCl2 is, however, less preferable due to causing environmental problems and contaminating the carbon products, which is a concern, especially in pharmaceutical and food industries [16] – this phrase should be replaced to line 129 (after …..with our previous work)
π → π* transition
(677) exposure to air during
Reviewer 3 Report
The manuscript has been sufficiently improved to warrant publication in Nanomaterials.
Author Response
Reviewer 3 report :The manuscript has been sufficiently improved to warrant publication in Nanomaterials.
Author Response:
We are grateful for your acceptance. We appreciate for your time and valuable feedback that make our paper complete considerably.
Best regards,
Parncheewa Udomsap